**Data Availability Statement:** All relevant data are within the paper and its Supporting Information files.

# Synergistic antibacterial effects of colistin in combination with aminoglycoside, carbapenems, cephalosporins, fluoroquinolones, tetracyclines, fosfomycin, and piperacillin on multidrug resistant *Klebsiella pneumoniae* isolates

Julalak C. Ontong[1,2,3], Nwabor F. Ozioma[1,2], Supayang P. Voravuthikunchai[2], Sarunyou Chusri[1] *

1 Division of Infectious Diseases, Department of Internal Medicine, Faculty of Medicine, Prince of Songkla University, Hat Yai, Songkhla, Thailand, 2 Division of Biological Science, Excellence Research Laboratory on Natural Products, Faculty of Science and Natural Product Research Center of Excellence, Prince of Songkla University, Hat Yai, Songkhla, Thailand, 3 Cosmetic Technology and Dietary Supplement Products Program, Faculty of Agro and Bio Industry, Thaksin University, Ban Pa Phayom, Phatthalung, Thailand

* sarunyouchusri@hotmail.com

## Abstract

Multidrug resistant Enterobacterales have become a serious global health problem, with extended hospital stay and increased mortality. Antibiotic monotherapy has been reported ineffective against most drug resistant bacteria including *Klebsiella pneumoniae*, thus encouraging the use of multidrug therapies as an alternative antibacterial strategy. The present works assessed the antibacterial activity of colistin against *K. pneumoniae* isolates. Resistant isolates were tested against 16 conventional antibiotics alone and in combination with colistin. The results revealed that all colistin resistant isolates demonstrated multidrug resistance against the tested antibiotics except amikacin. At sub-inhibitory concentrations, combinations of colistin with amikacin, or fosfomycin showed synergism against 72.72% (8 of 11 isolates). Colistin with either of gentamicin, meropenem, cefoperazone, cefotaxime, ceftazidime, moxifloxacin, minocycline, or piperacillin exhibited synergism against 81.82% (9 of 11 isolates). Combinations of colistin with either of tobramycin or ciprofloxacin showed synergism against 45.45% (5 in 11 isolates), while combinations of colistin with imipenem or ceftolozane and tazobactam displayed 36.36% (4 of 11 isolates) and 63.64% (7 of 11 isolates) synergism. In addition, combinations of colistin with levofloxacin was synergistic against 90.91% (10 of 11 isolates). The results revealed that combinations of colistin with other antibiotics could effectively inhibit colistin resistant isolates of *K. pneumoniae*, and thus could be further explore for the treatment of multidrug resistant pathogens.

**Funding:** This work was supported by Senior Research Scholar, the Thailand Research Fund, in the form of a grant awarded to SC (RTA6180006).

**Competing interests:** The authors declare that no competing interests exist.

## Introduction

The spread of multidrug resistant bacteria has become a public health emergency that threatens the continued usage of antibiotics chemotherapy. Infections due to carbapenemase-producing Enterobacterales are fast spreading across the globe, rendering the healthcare systems ineffective. It is estimated that infections associated with drug-resistant pathogens currently cause about 700,000 deaths annual, and if the status quo prevails could increase to 10 million annual deaths by 2050 [1]. *Klebsiella pneumoniae* is a Gram-negative Enterobacterales member commonly associated with hospital acquired infections. In the past, carbapenems were used as choice drugs for the treatment of *K. pneumoniae* infection. However, with the emergence and spread of carbapenemase-producing Enterobacterales, colistin is considered a last-resort treatment option for infections caused by carbapenem resistant bacterial isolates.

Colistin also known as polymyxin E is a cationic peptide that targets the negatively charged lipopolysaccharide (LPS) of Gram-negative bacteria. Colistin competitively displaces divalent cations ($Ca^{2+}$ and $Mg^{2+}$) from the phosphate groups of membrane lipids, resulting in the rupture of cell membrane and leakage of intracellular components [2]. Emergence of colistin resistance in Enterobacterales including *K. pneumoniae* has been reported in many parts of the globe [3, 4] and is fast spreading [5]. Colistin resistance was generally thought to be mediated by alterations and modification of the LPS target through addition of positively charged 4-amino-4-deoxy-L-arabinose (L-Ara4N) and phosphoethanolamine (pEtN) cationic molecules, responsible for decrease in bacterial outer membrane negative charge and reduced interaction with colistin [6]. However, recent scientific enquiries have reported the role of efflux pumps [7–10] and plasmid encoded *mcr* 1–9 genes [4, 11, 12] in colistin resistance.

As the fight against multidrug resistance continues, concerted efforts by agencies, health care systems and biomedical scientist are restlessly exploring possible alternatives that might suffice pending the discovery and development of novel antibacterial agents that could effectively curb the spread of antibiotics resistance. The use of antibiotics combinations [13–15], efflux pump inhibitors [16–18], and resistance modifying agents [19, 20] are suggested as temporary control measures to reverse microbial resistance or enhance the inactivation of resistant bacterial isolates. Antibiotics combination therapy is proposed as a reliable option with demonstrated results against multidrug resistant bacterial isolates.

The present study tested the antibiotic susceptibility of 85 *K. pneumoniae* isolates obtained from hospitals in the southern region of Thailand against colistin. The resistant isolates were further tested against several groups of antibiotics, and the synergistic antibacterial effects of combinations of colistin with other antibiotics were evaluated. The study presents *In vitro* experimental antibacterial data from broth micro-dilution technique and the checkerboard assay but did not monitor the time-kill kinetics.

## Materials and methods

### Chemicals and media

All culture media were purchased from Becton Dickinson & Co. Difco™ (Franklin Lakes, NJ, USA). Colistin sulfate, piperacillin, minocycline hydrochloride, tobramycin, and moxifloxacin hydrochloride were obtained from Sigma-Aldrich, (Saint Louis, MO, USA). Ciprofloxacin, cefotaxime, and levofloxacin were purchased from Siam Bheasach Co., Ltd. (Bangkok, Thailand). Tigecycline and ceftaroline fosamil were Pfizer Inc. (Philadelphia, PA, USA). Ceftazidime was obtained from Reyoung Pharmaceutical Co., Ltd. (Shandong, China). Imipenem was obtained from Merck Sharp & Dohme Corp. (Elkton, VA, USA). Meropenem was obtained from M&H manufacturing Co., Ltd. (Samutprakarn, Thailand). Cefoperazone and

sulbactam was obtained from L.B.S. Laboratory Ltd. (Bangkok, Thailand). Ceftolozane and tazobactam was obtained from Steri-Pharma, LLC (Syracuse, NY, USA). Fosfomycin was obtained from Meiji seikakaisna, Ltd. (Tokyo, Japan).

## Bacterial strains

Isolates collected from hospitalized patients unresponsive to antibiotics chemotherapy, with prolonged hospital stay in 8 hospitals located in Southern Thailand were used in this study. All isolates were identified to species level using standard biochemical tests and MALDI-TOF-MS. A total of 85 *K. pneumoniae* isolates that exhibited colony on MacConkey agar supplemented with 6 μg/mL imipenem were taken as resistant to carbapenems. The surveillance study was conducted post antibiotics treatments. *Escherichia coli* ATCC 25922 was used as a quality control. All the bacterial cultures were stored in tryptic soy broth (TSB supplemented with 40% glycerol and kept at -80°C.

## Antimicrobial susceptibility testing

Minimum inhibitory concentrations (MICs) were determined by the broth microdilution method in according to Clinical and Laboratory Standards Institute (CLSI) guidelines [21]. Briefly, serial two-fold dilutions of antibiotics were prepared in cation-adjusted Mueller-Hinton II broth (CA-MHBII). To investigate the effect of each antibiotic, an aliquot of 100 μL of diluted bacterial suspension ($1 \times 10^6$ CFU/ml) was mixed with 100 μL antibiotic into each well and incubated at 37°C for 18 h. MIC was expressed as the lowest concentration of the antibiotic that inhibits visible growth after incubation. Following European Committee on Antimicrobial Susceptibility Testing (EUCAST) breakpoints, isolates with a colistin MIC $\leq$ 2 μg/mL were categorized as susceptible and those with a colistin MIC $\geq$ 4 μg/mL were categorized as resistant [22]. To determine the MBC, MIC and supra-MIC dilutions were spotted on an agar plate and incubated overnight at 37°C. Bacterial growth was observed, and MBC was defined as the lowest concentration that showed no visible bacterial regrowth.

## Checkerboard technique

The synergistic effects of colistin and any of sixteen other antibiotics (amikacin, gentamicin, tobramycin, imipenem, meropenem, cefoperazone, cefotaxime, ceftazidime, ceftolozane and tazobactam, ciprofloxacin, levofloxacin, moxifloxacin, minocycline, tigecycline, fosfomycin, and piperacillin) on *K. pneumoniae* isolates were tested using the checkerboard technique. Each antibiotic was diluted to concentrations ranging from 1/64 MIC to 8 MIC of the previously determined MIC. Briefly, 50 μL of colistin and each antibiotic were assessed by adding 100 μL diluted bacterial suspension ($1 \times 10^6$ CFU/mL) into a well containing 50 μL colistin and 50 μL one of sixteen other antibiotics. The test was then read after 18 h of incubation at 37 C. Each value was mean of triplicates from three independent experiments. The effects of the antimicrobial combinations were defined according to the fractional inhibitory concentration index (FICI) as following equation:

$$\text{FICI} = \frac{\text{MIC of drug A in combination}}{\text{MIC of drug A alone}} + \frac{\text{MIC of drug B in combination}}{\text{MIC of drug B alone}}$$

The FICI results for each combination were interpreted as follows: FICI $\leq$ 0.5, synergism; 0.5 < FICI < 1, additive; 1 $\leq$ FICI < 2, indifference; and FICI $\geq$ 2, antagonism. *Escherichia coli* ATCC 25922 was used as standard control strains for the assays [23].

### Ethical statement

This retrospective study was approved by the Institutional Review Board (IRB) of the Faculty of Medicine, Prince of Songkla University, Thailand (EC: 54-080-14-1-2.). The researchers were granted permission to extract the data from the database with waiver of consent because of the observational nature of the study. All data were fully anonymized before the researcher accessed and analyzed.

## Results and discussion

### Distribution of *K. pneumoniae* isolates

In this study *K. pneumoniae* isolates obtained from patients receiving treatment in tertiary hospitals were tested against colistin. The isolates were collected and tested for antibiotics susceptibility due to patient's unresponsiveness to antibiotics treatments and suspicion of opportunistic role of drug-resistant bacterial colonizers in aggravating health conditions of immunocompromised patients. Table 1. shows the distribution of the isolates, collected from eight public hospitals located in the southern region of Thailand based on sample type and location of the hospital. The results revealed that all nine isolates from Narathiwat hospital were resistant to colistin, while one out of the three isolates from Songkhla and the only isolate from Trang were also resistant to colistin. Demographic data, clinical characteristics, and outcomes of the patients with colonization due to colistin-resistant and carbapenem-resistant *K. pneumoniae* are presented in S1 Table. The samples were collected from adult patients between the ages of 25–94 years, who were admitted in the ICU units of the hospitals. Most of the patients had underlining health conditions including diabetes mellitus, hypertension, dyslipidemia, coronary artery disease, cerebrovascular disease, chronic kidney disease, and chronic obstructive pulmonary disease.

### Antibacterial activities of colistin against *K. pneumoniae* isolates

A total of 85 isolates were tested for susceptibility to colistin using broth microdilution assay (Table 2 and S2 Table). The results showed that 74 isolates (87.05%) were susceptible to colistin with MIC values ranging from 0.5 to 2.0 μg/mL and MBC values ranging from 0.5 to 4.0 μg/mL. However, 11 isolates (12.94%) demonstrated resistance to colistin with MIC and MBC values ranging from 256 to >1024 μg/mL and 512 to >1024 μg/mL, respectively. Colistin resistant *K. pneumoniae* has been reported in various countries and regions including Netherlands, America, Nigeria as well as Thailand [24–26]. Recently 213 of 280 (79.1%) *K. pneumoniae* isolates obtained from humans in Thailand showed colistin resistance [27]. The emergence of colistin resistance *K. pneumoniae* presents a major threat to public health, since colistin represents the last line drug of choice against carbapenem resistant *K. pneumoniae*.

**Table 1. Distribution of *K. pneumoniae* isolates based on hospital location and sample type.**

| Hospitals | Gastric content | Throat | Rectal | Endotracheal tube | Environmental | Total resistant |
|---|---|---|---|---|---|---|
| Hat Yai | 4 | 6 | 6 | 4 | 0 | 0 |
| Narathiwat | 3(R) | 2(R) | 3(R) | 1(R) | 0 | 9 |
| Pattani | 5 | 5 | 8 | 3 | 1 | 0 |
| Phatthalung | 4 | 5 | 4 | 1 | 2 | 0 |
| Songklanagarind | 2 | 1 | 4 | 0 | 0 | 0 |
| Songkhla | 2 | 0 | 1(R) | 0 | 0 | 1 |
| Satun | 2 | 1 | 4 | 0 | 0 | 0 |
| Trang | 0 | 0 | 1(R) | 0 | 0 | 1 |
| Total | 22 | 20 | 31 | 9 | 3 | 11 |

**Table 2. Minimum inhibitory and minimum bactericidal concentrations of colistin on *K. pneumoniae* isolates.**

| Isolates (n) | MIC | MBC |
|---|---|---|
| Susceptible (74) | 0.5–2.0 | 0.5–4.0 |
| Resistant (11) | 256 –>1025 | 512 –>1025 |

## Antibiogram of colistin resistant *K. pneumoniae* isolates

The eleven resistant isolates were evaluated for antibiogram against sixteen conventional antibiotics and colistin (Table 3). Although the agar dilution method is recommended as reference method for the determination of Fosfomycin MICs, using the reference agar dilution method in a checkerboard analysis is practically difficult. Thus, the broth microdilution with glucose-6-phosphate (G-6-P) was used [28]. Results of the assay were interpreted based on CLSI and EUCAST breakpoint standards [22, 29]. All colistin resistant isolates exhibited multi-drug resistance to aminoglycosides, carbapenems, cephalosporins, fluoroquinolones, fosfomycin, tetracyclines, and piperacillin, but were susceptible to amikacin with MIC ranging from 4 to 16 µg/mL. Previous researchers have reported 69.57 and 64.1% susceptibility of *K. pneumoniae* to amikacin [30, 31]. Resistance to carbapenems (imipenem and meropenem) was observed for all colistin resistant isolates. Two isolates (1SK5R and 1TR5R) were susceptible to cefoperazone, ceftolozane and tazobactam, levofloxacin, fosfomycin, and piperacillin. In addition, isolate 1TR5R was susceptible to tobramycin, ceftazidime, and minocycline. Multidrug resistant *K. pneumoniae* has been reported by previous researchers [32, 33], and is globally spreading unabated. This might lead to increased hospital stay and *K. pneumoniae* associated mortality.

## Synergistic effects of colistin-antibiotics combinations

While the search for alternative antimicrobial agents that can effectively control the spread of multidrug resistance continues, various temporary measures are been employed for the treatment of infections caused by drug resistant bacterial isolates. Antibiotic combination therapy

**Table 3. Minimum inhibitory concentrations of colistin resistant *K. pneumoniae* isolates against conventional antibiotics.**

| Isolates | Antibiotics (µg/mL) | | | | | | | | | | | | | | | | |
|---|---|---|---|---|---|---|---|---|---|---|---|---|---|---|---|---|---|
| | Aminoglycosides | | | | | Carbapenem | | Cephalosporins | | | | | Fluoroquinolones | Tetracycline | | | |
| | | | | | | | | | | | | | | | | Fosfomycin | Penicillin |
| COL | AMI | GEN | TOB | IMI | | MER | CEFZ | CEFX | CEFD | CEFT | CIP | LEV | MOX | MIN | TIG | FOS | PIP |
| 1NT4Ng/1 | 256 | 16 | 128 | 32 | 256 | 256 | >1024 | >1024 | >1024 | >1024 | >256 | 512 | 512 | 128 | 8 | >1024 | >1024 |
| 1NT6Ng/1 | 256 | 16 | 128 | 32 | 256 | 256 | >1024 | >1024 | >1024 | >1024 | >256 | 512 | 512 | 128 | 8 | >1024 | >1024 |
| 1NT6Tu/1 | 256 | 8 | 256 | 64 | 128 | 256 | >1024 | >1024 | >1024 | >1024 | >256 | 512 | 512 | 128 | 8 | >1024 | >1024 |
| 1NT6R | 256 | 16 | 128 | 64 | 128 | 256 | >1024 | >1024 | >1024 | >1024 | >256 | 512 | 512 | 128 | 8 | >1024 | >1024 |
| 1NT7R | 256 | 8 | 128 | 32 | 256 | 256 | >1024 | >1024 | >1024 | >1024 | >256 | 512 | 512 | 128 | 8 | >1024 | >1024 |
| 1NT8Th | 256 | 16 | 64 | 32 | 128 | 128 | >1024 | >1024 | >1024 | >1024 | >256 | 512 | 512 | 128 | 8 | >1024 | >1024 |
| 1NT6Ng (CCU)/1 | 512 | 16 | 128 | 16 | 128 | 256 | >1024 | >1024 | >1024 | >1024 | 128 | 64 | 32 | 32 | 8 | 128 | >1024 |
| 1NT6Th(CCU)/1 | 256 | 16 | 128 | 16 | 128 | 128 | >1024 | >1024 | >1024 | >1024 | 128 | 64 | 32 | 32 | 8 | 128 | >1024 |
| 1NT6R(CCU) | 256 | 16 | 128 | 32 | 128 | 128 | >1024 | >1024 | >1024 | >1024 | 256 | 512 | 512 | 128 | 8 | >1024 | >1024 |
| 1SK5R | >1024 | 4 | 4 | 8 | 128 | 256 | 4 | 512 | 256 | 4 | 32 | 0.50 | 4 | 16 | 16 | 32 | 16 |
| 1TR5R | >1024 | 4 | 0.50 | 1 | 32 | 32 | 2 | 4 | 0.25 | 1 | 4 | 0.25 | 4 | 0.25 | 8 | 0.5 | 0.5 |

AMI, Amikacin; CCU, Cardiac Care Unit; CEFD, Ceftazidime; CEFT, Ceftolozane and Tazobactam; CEFX, Cefotaxime; CEFZ, Cefoperazone; CIP, Ciprofloxacin; COL, Colistin; FOS, Fosfomycin; GEN, Gentamicin; IMI, Imipenem; LEV, Levofloxacin; MER, Meropenem; MIN, Minocycline; MOX, Moxifloxacin; PIP, Piperacillin; TIG, Tigecycline; TOB, Tobramycin.

is a possible effective option which currently is attracting numerous research attention [34–36]. Combinations of antibiotics effectively inhibit microbial proliferation through a multi-target approach resulting in the microbial death. Hence, the synergistic effects of colistin with conventional antibiotics (amikacin, gentamicin, tobramycin, imipenem, meropenem, cefoperazone, cefotaxime, ceftazidime, ceftolozane and tazobactam, ciprofloxacin, levofloxacin, moxifloxacin, minocycline, tigecycline, fosfomycin, and piperacillin) against *K. pneumoniae* isolates were evaluated and classified based on FICI parameter (S3 Table).

## Combination of colistin with aminoglycosides

Table 4 presents the chequerboard results of colistin in combination with aminoglycoside (amikacin, gentamicin, and tobramycin) against the eleven colistin resistant *K. pneumoniae* isolates. The FICI ranged from 0.125 to 0.500 for *K. pneumoniae* isolates, except for isolates 1NT6R and 1NT6Th (CCU)/1 with amikacin FICI values of 0.078 and 0.062, respectively. However, combination of colistin with amikacin or tobramycin showed no synergy against isolate 1NT7R. Antibacterial synergistic effects have been demonstrated for gentamicin and amikacin combinations with polymyxin B and ceftazidime-avibactam [37, 38].

## Combination of colistin with carbapenems

The antibacterial activities of colistin in combination with carbapenems (imipenem and meropenem) is presented in Table 5. The results demonstrated synergism with FICI values ranging from 0.250 to 0.500 for most isolates, except isolates 1SK5R and 1TR5R. Combinations of colistin and imipenem however showed no effects on isolates 1NT6Tu/1, 1NT8Th, 1NT6Ng (CCU/1), 1NT6Th (CCU)/1, and 1NT6R (CCU). The antibacterial activities of fosfomycin and meropenem combination and colistin with meropenem combinations effective inhibited NDM and carbapenemase producing *K. pneumoniae* [34, 39]. In addition, relebactam-imipenem combinations showed enhanced antibacterial activity against colistin resistant *K. pneumoniae* with potentials of restoring bacteria susceptibility to imipenem [40, 41].

## Combination of colistin with cephalosporins

Colistin–cephalosporins (Cefoperazone, Cefotaxime, Ceftazidime and Ceftolozane and Tazobactam) combinations were also assessed for synergism against colistin resistant *K. pneumoniae* isolates Table 6. Combinatory effect of colistin with cephalosporins revealed FICIs ranging from 0.187 to 0.375 for all isolates, except 1SK5R and 1TR5R, while combination with ceftolozane and tazobactam were ineffective against 1NT8Th and 1NT6R (CCU). The results suggested collaborative disruption of the cell wall since both colistin and cephalosporins targets strategic components of the cell wall. Ceftazidime-avibactam in combination with colistin was previous reported to be effective against colistin non-susceptible strains of multidrug resistant (MDR) *Pseudomonas aeruginosa* [42, 43]. In addition, combinations of ceftazidime/avibactam and colistin, tobramycin, or tigecycline were effective against OXA-48-producing *Enterobacterales* strains [44].

## Combination of colistin with fluoroquinolones

The effects of colistin combination with fluoroquinolones (ciprofloxacin, levofloxacin, moxifloxacin) are presented in Table 7. The FICIs (0.093 to 0.500) indicated synergistic effects against most isolates. However, combinations of colistin with all three antibiotics were ineffective on isolate 1TR5R, while ciprofloxacin or moxifloxacin were also not effective on isolate 1SK5R.

**Table 4. Chequerboard results of colistin in combination with aminoglycoside (amikacin, gentamicin and tobramycin) against colistin resistant *K. pneumoniae* isolates.**

| Isolates | Antibiotics | Combined MIC (µg/mL) | FICI | Fold reduction | Outcome |
|---|---|---|---|---|---|
| 1NT4Ng/1 | COL | 8 | 0.281 | 32 | Synergy |
|  | AMI | 4 |  | 4 |  |
|  | COL | 32 | 0.375 | 8 | Synergy |
|  | GEN | 32 |  | 4 |  |
|  | COL | 64 | 0.500 | 4 | Synergy |
|  | TOB | 8 |  | 4 |  |
| 1NT6Ng/1 | COL | 8 | 0.281 | 32 | Synergy |
|  | AMI | 4 |  | 4 |  |
|  | COL | 8 | 0.281 | 32 | Synergy |
|  | GEN | 32 |  | 4 |  |
|  | COL | 64 | 0.500 | 4 | Synergy |
|  | TOB | 8 |  | 4 |  |
| 1NT6Tu/1 | COL | 64 | 0.265 | 4 | Synergy |
|  | AMI | 0.12 |  | 64 |  |
|  | COL | 8 | 0.281 | 32 | Synergy |
|  | GEN | 64 |  | 4 |  |
|  | COL | ND | ND | ND | ND |
|  | TOB | ND |  |  |  |
| 1NT6R | COL | 16 | 0.078 | 16 | Synergy |
|  | AMI | 0.25 |  | 64 |  |
|  | COL | 16 | 0.125 | 16 | Synergy |
|  | GEN | 8 |  | 16 |  |
|  | COL | 32 | 0.375 | 8 | Synergy |
|  | TOB | 16 |  | 4 |  |
| 1NT7R | COL | ND | ND | ND | ND |
|  | AMI | ND |  |  |  |
|  | COL | 32 | 0.375 | 8 | Synergy |
|  | GEN | 32 |  | 4 |  |
|  | COL | ND | ND | ND | ND |
|  | TOB | ND |  |  |  |
| 1NT8Th | COL | 16 | 0.312 | 16 | Synergy |
|  | AMI | 4 |  | 4 |  |
|  | COL | 64 | 0.500 | 4 | Synergy |
|  | GEN | 16 |  | 4 |  |
|  | COL | ND | ND | ND | ND |
|  | TOB | ND |  |  |  |
| 1NT6Ng(CCU)/1 | COL | 16 | 0.281 | 32 | Synergy |
|  | AMI | 4 |  | 4 |  |
|  | COL | 64 | 0.250 | 8 | Synergy |
|  | GEN | 16 |  | 8 |  |
|  | COL | 64 | 0.375 | 8 | Synergy |
|  | TOB | 4 |  | 4 |  |
| 1NT6Th(CCU)/1 | COL | 8 | 0.062 | 32 | Synergy |
|  | AMI | 0.5 |  | 32 |  |
|  | COL | 32 | 0.375 | 8 | Synergy |
|  | GEN | 32 |  | 4 |  |
|  | COL | 16 | 0.312 | 16 | Synergy |
|  | TOB | 4 |  | 4 |  |

(*Continued*)

**Table 4.** (Continued)

| Isolates | Antibiotics | Combined MIC (µg/mL) | FICI | Fold reduction | Outcome |
|---|---|---|---|---|---|
| 1NT6R(CCU) | COL | 32 | 0.250 | 8 | Synergy |
| | AMI | 2 | | 8 | |
| | COL | 32 | 0.187 | 8 | Synergy |
| | GEN | 8 | | 16 | |
| | COL | ND | ND | ND | ND |
| | TOB | | | | |

CCU, Cardiac Care Unit; MIC, Minimum inhibitory concentrations; COL, Colistin; AMI, Amikacin; FICI, Fractional inhibitory concentration index; GEN, Gentamicin; TOB, Tobramycin; ND, not determined.

### Combination of colistin with fosfomycin, tetracyclines (minocycline, and tigecycline) and penicillin (piperacillin)

In addition, effects of colistin in combination with fosfomycin, tetracyclines (minocycline, and tigecycline) and penicillin (piperacillin) resistant *K. pneumoniae* isolates is presented in Table 8. Combination of colistin with fosfomycin showed FICI ranging from 0.062 to 0.500 against most isolates except isolates 1NT6Tu/1, 1NT6R, and 1SK5R while combination of colistin with piperacillin range of 0.046 to 0.375 for most isolates except 1NT7R and 1SK5R. Against all isolates, combination of colistin with tigecycline was not synergistic, whereas combinations with minocycline demonstrated synergistic activity against most isolates with FICI value of 0.093, except for isolate 1NT6R (CCU) with FICI value of 0.062, and were not effective against 1SK5R and 1TR5R.

## Discussion

The rapid spread of multidrug resistant pathogenic bacteria places an enormous responsibility on global health care systems. Moreover, the search and development of new antibiotics is far too slow with no novel discovery over the last 30 years. This shortage of effective therapeutic agents has encouraged trial of combinations of existing agents for synergistic activities against drug resistant isolates. The present study combined colistin a last line drug for multidrug resistant Enterobacterales with 16 other antibiotics that are not effective against *K. pneumoniae*. A total of 11 isolates (12.94%) presented resistance to colistin. Nine out of the 11 resistant isolates were obtained from Narathiwat hospital, with a 100% resistance prevalence. The results suggested high prevalence of resistance amongs *K. pneumoniae* in Narathiwat hospital which might be due to local tranmission within the hospital. Furthermore, 1 among 3 isolates obtained from Songkhla hospital was resistant, whereas the single isolate obtained from Trang hospital was also resistant. However, due to the few number of isolates used in the study, epidemiological statements on the prevalence of colistin resistant isolates in this regions might be baised. Colistin resistant *K. pneumoniae* has previously been reported in various countries and regions including Netherlands, America, Nigeria as well as Thailand [24–26]. Recently 213 of 280 (79.1%) *K. pneumoniae* isolates obtained from humans in Thailand showed colistin resistance [27].

All colistin–resistant isolates were also resistant to other antibiotics including aminoglycosides, carbapenems, cephalosporins, fluoroquinolones, fosfomycin, tetracyclines, and piperacillin, but were susceptible to amikacin with MIC ranging from 4 to 16 µg/mL. Previous researchers have reported 69.57 and 64.1% susceptibility of *K. pneumoniae* to amikacin [30, 31]. The results suggested that amikacin might be a drug option in the management of drug

**Table 5. Chequerboard results of colistin in combination with carbapenems (imipenem and meropenem) against colistin resistant *K. pneumoniae* isolates.**

| Isolates | Antibiotics | Combined MIC (μg/mL) | FICI | Fold reduction | Outcome |
|---|---|---|---|---|---|
| 1NT4Ng/1 | COL | 64 | 0.375 | 4 | Synergy |
| | IMI | 32 | | 8 | |
| | COL | 64 | 0.375 | 4 | Synergy |
| | MER | 32 | | 8 | |
| 1NT6Ng/1 | COL | 64 | 0.500 | 4 | Synergy |
| | IMI | 64 | | 4 | |
| | COL | 64 | 0.375 | 4 | Synergy |
| | MER | 32 | | 8 | |
| 1NT6Tu/1 | COL | ND | ND | ND | ND |
| | IMI | ND | | | |
| | COL | 16 | 0.312 | 16 | Synergy |
| | MER | 64 | | 4 | |
| 1NT6R | COL | 64 | 0.500 | 4 | Synergy |
| | IMI | 32 | | 4 | |
| | COL | 16 | 0.312 | 16 | Synergy |
| | MER | 64 | | 4 | |
| 1NT7R | COL | 64 | 0.281 | 4 | Synergy |
| | IMI | 4 | | 64 | |
| | COL | 32 | 0.250 | 8 | Synergy |
| | MER | 32 | | 8 | |
| 1NT8Th | COL | ND | ND | ND | ND |
| | IMI | ND | | | |
| | COL | 64 | 0.500 | 4 | Synergy |
| | MER | 32 | | 4 | |
| 1NT6Ng(CCU)/1 | COL | ND | ND | ND | ND |
| | IMI | ND | | | |
| | COL | 32 | 0.312 | 16 | Synergy |
| | MER | 64 | | 4 | |
| 1NT6Th(CCU)/1 | COL | ND | ND | ND | ND |
| | IMI | ND | | | |
| | COL | 64 | 0.500 | 4 | Synergy |
| | MER | 32 | | 4 | |
| 1NT6R(CCU) | COL | ND | ND | ND | ND |
| | IMI | ND | | | |
| | COL | 64 | 0.500 | 4 | Synergy |
| | MER | 32 | | 4 | |

CCU, Cardiac Care Unit; MIC, Minimum inhibitory concentrations; COL, Colistin; FICI, Fractional inhibitory concentration index; IMI, Imipenem; MER, Meropenem; MIC, ND, not determined.

resistant Enterobacterales, and should be further explored. Resistance to carbapenems (imipenem and meropenem) was observed for all colistin resistant isolates. Two isolates (1SK5R and 1TR5R) were susceptible to cefoperazone, ceftolozane and tazobactam, levofloxacin, fosfomycin, and piperacillin. In addition, isolate 1TR5R was susceptible to tobramycin, ceftazidime, and minocycline. Multidrug resistance in *K. pneumoniae* has been reported by previous researchers [32, 33], resulting in increased hospital stay and *K. pneumoniae* associated mortality.

**Table 6. Chequerboard results of colistin in combination with cephalosporins (cefoperazone, cefotaxime, ceftazidime and ceftolozane and tazobactam) against colistin resistant *K. pneumoniae* isolates.**

| Isolates | Antibiotics | Combined MIC (µg/mL) | FICI | Fold reduction | Outcome |
|---|---|---|---|---|---|
| 1NT4Ng/1 | COL | 32 | 0.375 | 8 | Synergy |
| | CEFZ | 256 | | 4 | |
| | COL | 16 | 0.312 | 16 | Synergy |
| | CEFX | 256 | | 4 | |
| | COL | 16 | 0.312 | 16 | Synergy |
| | CEFD | 256 | | 4 | |
| | COL | 64 | 0.375 | 4 | Synergy |
| | CEFT | 128 | | 8 | |
| 1NT6Ng/1 | COL | 32 | 0.375 | 8 | Synergy |
| | CEFZ | 256 | | 4 | |
| | COL | 16 | 0.312 | 16 | Synergy |
| | CEFX | 256 | | 4 | |
| | COL | 32 | 0.375 | 8 | Synergy |
| | CEFD | 256 | | 4 | |
| | COL | 64 | 0.253 | 4 | Synergy |
| | CEFT | 16 | | 64 | |
| 1NT6Tu/1 | COL | 16 | 0.312 | 16 | Synergy |
| | CEFZ | 256 | | 4 | |
| | COL | 16 | 0.312 | 16 | Synergy |
| | CEFX | 256 | | 4 | |
| | COL | 16 | 0.312 | 16 | Synergy |
| | CEFD | 256 | | 4 | |
| | COL | 32 | 0.187 | 8 | Synergy |
| | CEFT | 64 | | 16 | |
| 1NT6R | COL | 16 | 0.312 | 16 | Synergy |
| | CEFZ | 256 | | 4 | |
| | COL | 16 | 0.312 | 16 | Synergy |
| | CEFX | 256 | | 4 | |
| | COL | 16 | 0.312 | 16 | Synergy |
| | CEFD | 256 | | 4 | |
| | COL | 64 | 0.257 | 4 | Synergy |
| | CEFT | 8 | | 128 | |
| 1NT7R | COL | 16 | 0.312 | 16 | Synergy |
| | CEFZ | 256 | | 4 | |
| | COL | 16 | 0.312 | 16 | Synergy |
| | CEFX | 256 | | 4 | |
| | COL | 32 | 0.250 | 8 | Synergy |
| | CEFD | 128 | | 8 | |
| | COL | 64 | 0.257 | 4 | Synergy |
| | CEFT | 8 | | 128 | |
| 1NT8Th | COL | 64 | 0.281 | 4 | Synergy |
| | CEFZ | 32 | | 32 | |
| | COL | 16 | 0.312 | 16 | Synergy |
| | CEFX | 256 | | 4 | |
| | COL | 16 | 0.187 | 16 | ND |
| | CEFD | 128 | | 8 | |
| | COL | ND | ND | ND | ND |
| | CEFT | ND | | | |

*(Continued)*

**Table 6.** (Continued)

| Isolates | Antibiotics | Combined MIC (µg/mL) | FICI | Fold reduction | Outcome |
|---|---|---|---|---|---|
| 1NT6Ng(CCU)/1 | COL | 64 | 0.375 | 8 | Synergy |
| | CEFZ | 256 | | 4 | |
| | COL | 16 | 0.281 | 32 | Synergy |
| | CEFX | 256 | | 8 | |
| | COL | 64 | 0.187 | 8 | Synergy |
| | CEFD | 64 | | 16 | |
| | COL | 128 | 0.265 | 4 | Synergy |
| | CEFT | 16 | | 64 | |
| 1NT6Th(CCU)/1 | COL | 16 | 0.312 | 16 | Synergy |
| | CEFZ | 256 | | 4 | |
| | COL | 16 | 0.312 | 16 | Synergy |
| | CEFX | 256 | | 4 | |
| | COL | 32 | 0.250 | 8 | Synergy |
| | CEFD | 128 | | 8 | |
| | COL | 64 | 0.281 | 4 | Synergy |
| | CEFT | 32 | | 32 | |
| 1NT6R(CCU) | COL | 16 | 0.312 | 16 | Synergy |
| | CEFZ | 256 | | 4 | |
| | COL | 32 | 0.250 | 8 | Synergy |
| | CEFX | 128 | | 8 | |
| | COL | 16 | 0.312 | 16 | Synergy |
| | CEFD | 256 | | 4 | |
| | COL | ND | ND | ND | ND |
| | CEFT | ND | | | |

CCU, Cardiac Care Unit; MIC, Minimum inhibitory concentrations; COL, Colistin; CEFD, Ceftazidime; CEFT, Ceftolozane and Tazobactam; CEFX, Cefotaxime; CEFZ, Cefoperazone; FICI, Fractional inhibitory concentration index; ND, not determined.

While the search for alternative and effective antimicrobial agents continues, various temporary measures are been employed for the treatment of infections caused by drug resistant bacterial isolates. Antibiotic combination therapy is a possible effective option which currently is attracting numerous research attention [34–36]. Combinations of antibiotics effectively inhibit microbial proliferation through a multi-target approach resulting in effective inactivation of cells and microbial death. Hence, the synergistic effects of colistin with conventional antibiotics (amikacin, gentamicin, tobramycin, imipenem, meropenem, cefoperazone, cefotaxime, ceftazidime, ceftolozane and tazobactam, ciprofloxacin, levofloxacin, moxifloxacin, minocycline, tigecycline, fosfomycin, and piperacillin) against multi-drug resistant *K. pneumoniae* isolates were evaluated and classified based on FICI parameter (S3 Table). Antibiotic combination demonstrated 5- to 64-fold reduction in MIC of colistin and 4-to 512-fold reduction in MIC of tested antibiotics. A summary of the results indicates that colistin with amikacin, or fosfomycin combinations were synergistic against 72.72% (8 of 11 isolates). Colistin with gentamicin, or meropenem, or cefoperazone, or cefotaxime, or ceftazidime, or moxifloxacin, or minocycline, or piperacillin exhibited synergism against 81.82% (9 of 11 isolates). Combinations of colistin with either of tobramycin or ciprofloxacin showed 45.45% (5 in 11 isolates), while combinations of colistin with imipenem or ceftolozane and tazobactam displayed 36.36% (4 of 11 isolates) and 63.64% (7 of 11 isolates) synergism. In addition, combinations of

**Table 7. Chequerboard results of colistin in combination with fluoroquinolones (ciprofloxacin, levofloxacin and moxifloxacin) against colistin resistant *K. pneumoniae* isolates.**

| Isolates | Antibiotics | Combined MIC (μg/mL) | FICI | Fold reduction | Outcome |
|---|---|---|---|---|---|
| 1NT4Ng/1 | COL | 64 | 0.257 | 4 | Synergy |
| | CIP | 2 | | 128 | |
| | COL | 16 | 0.125 | 16 | Synergy |
| | LEV | 32 | | 16 | |
| | COL | 8 | 0.093 | 32 | Synergy |
| | MOX | 32 | | 16 | |
| 1NT6Ng/1 | COL | 64 | 0.257 | 4 | Synergy |
| | CIP | 2 | | 128 | |
| | COL | 16 | 0.187 | 16 | Synergy |
| | LEV | 64 | | 8 | |
| | COL | 8 | 0.093 | 32 | Synergy |
| | MOX | 32 | | 16 | |
| 1NT6Tu/1 | COL | ND | ND | ND | ND |
| | CIP | ND | | | |
| | COL | 16 | 0.093 | 16 | Synergy |
| | LEV | 16 | | 32 | |
| | COL | 8 | 0.093 | 32 | Synergy |
| | MOX | 32 | | 16 | |
| 1NT6R | COL | 32 | 0.375 | 8 | Synergy |
| | CIP | 64 | | 4 | |
| | COL | 16 | 0.125 | 16 | Synergy |
| | LEV | 32 | | 16 | |
| | COL | 8 | 0.093 | 32 | Synergy |
| | MOX | 32 | | 16 | |
| 1NT7R | COL | 64 | 0.500 | 4 | Synergy |
| | CIP | 64 | | 4 | |
| | COL | 16 | 0.125 | 16 | Synergy |
| | LEV | 32 | | 16 | |
| | COL | 16 | 0.125 | 16 | Synergy |
| | MOX | 32 | | 16 | |
| 1NT8Th | COL | 64 | 0.500 | 4 | Synergy |
| | CIP | 64 | | 4 | |
| | COL | 16 | 0.125 | 16 | Synergy |
| | LEV | 32 | | 16 | |
| | COL | 16 | 0.093 | 16 | Synergy |
| | MOX | 16 | | 32 | |
| 1NT6Ng(CCU)/1 | COL | ND | ND | ND | ND |
| | CIP | ND | | | |
| | COL | 64 | 0.250 | 8 | Synergy |
| | LEV | 8 | | 8 | |
| | COL | 128 | 0.312 | 4 | Synergy |
| | MOX | 2 | | 16 | |
| 1NT6Th(CCU)/1 | COL | ND | ND | ND | ND |
| | CIP | ND | | | |
| | COL | 32 | 0.250 | 8 | Synergy |
| | LEV | 8 | | 8 | |
| | COL | 32 | 0.156 | 8 | Synergy |
| | MOX | 1 | | 32 | |

(*Continued*)

**Table 7.** (Continued)

| Isolates | Antibiotics | Combined MIC (μg/mL) | FICI | Fold reduction | Outcome |
|---|---|---|---|---|---|
| 1NT6R(CCU) | COL | ND | ND | ND | ND |
| | CIP | ND | | | |
| | COL | 16 | 0.187 | 16 | Synergy |
| | LEV | 64 | | 8 | |
| | COL | 8 | 0.093 | 32 | Synergy |
| | MOX | 32 | | 16 | |
| 1SK5R | COL | ND | ND | ND | ND |
| | CIP | ND | | | |
| | COL | 16 | 0.255 | 64 | Synergy |
| | LEV | 0.12 | | 4 | |
| | COL | ND | ND | ND | ND |
| | MOX | ND | | | |

CCU, Cardiac Care Unit; MIC, Minimum inhibitory concentrations; CIP, Ciprofloxacin; COL, Colistin; FICI, Fractional inhibitory concentration index; LEV, Levofloxacin; MOX, Moxifloxacin; ND, not determined.

**Table 8. Chequerboard results of colistin in combination with fosfomycin, tetracyclines (minocycline, and tigecycline) and penicillin (piperacillin) against colistin resistant *K. pneumoniae* isolates.**

| Isolates | Antibiotics | Combined MIC (μg/mL) | FICI | Fold reduction | Outcome |
|---|---|---|---|---|---|
| 1NT4Ng/1 | COL | 64 | 0.500 | 4 | Synergy |
| | FOS | 256 | | 4 | |
| | COL | 16 | 0.093 | 16 | Synergy |
| | MIN | 4 | | 32 | |
| | COL | ND | ND | ND | ND |
| | TIG | ND | | | |
| | COL | 32 | 0.128 | 8 | Synergy |
| | PIP | 4 | | 256 | |
| 1NT6Ng/1 | COL | 64 | 0.375 | 4 | Synergy |
| | FOS | 128 | | 8 | |
| | COL | 16 | 0.093 | 16 | Synergy |
| | MIN | 4 | | 32 | |
| | COL | ND | ND | ND | ND |
| | TIG | ND | | | |
| | COL | 32 | 0.187 | 8 | Synergy |
| | PIP | 64 | | 16 | |
| 1NT6Tu/1 | COL | ND | ND | ND | ND |
| | FOS | ND | | | |
| | COL | 16 | 0.093 | 16 | Synergy |
| | MIN | 4 | | 32 | |
| | COL | ND | ND | ND | ND |
| | TIG | ND | | | |
| | COL | 32 | 0.126 | 8 | Synergy |
| | PIP | 2 | | 512 | |

(*Continued*)

**Table 8.** (Continued)

| Isolates | Antibiotics | Combined MIC (µg/mL) | FICI | Fold reduction | Outcome |
|---|---|---|---|---|---|
| 1NT6R | COL | ND | ND | ND | ND |
| | FOS | ND | | | |
| | COL | 16 | 0.093 | 16 | Synergy |
| | MIN | 4 | | 32 | |
| | COL | ND | ND | ND | ND |
| | TIG | ND | | | |
| | COL | 64 | 0.251 | 4 | Synergy |
| | PIP | 2 | | 512 | |
| 1NT7R | COL | 64 | 0.500 | 4 | Synergy |
| | FOS | 256 | | 4 | |
| | COL | 16 | 0.093 | 16 | Synergy |
| | MIN | 4 | | 32 | |
| | COL | ND | ND | ND | ND |
| | TIG | ND | | | |
| | COL | ND | ND | ND | ND |
| | PIP | ND | | | |
| 1NT8Th | COL | 64 | 0.500 | 4 | Synergy |
| | FOS | 256 | | 4 | |
| | COL | 16 | 0.093 | 16 | Synergy |
| | MIN | 4 | | 32 | |
| | COL | ND | ND | ND | ND |
| | TIG | ND | | ND | |
| | COL | 64 | 0.251 | 4 | Synergy |
| | PIP | 2 | | 512 | |
| 1NT6Ng(CCU)/1 | COL | 32 | 0.078 | 16 | Synergy |
| | FOS | 2 | | 64 | |
| | COL | 16 | 0.093 | 32 | Synergy |
| | MIN | 2 | | 16 | |
| | COL | ND | ND | ND | ND |
| | TIG | ND | | | |
| | COL | 16 | 0.046 | 32 | Synergy |
| | PIP | 16 | | 64 | |
| 1NT6Th(CCU)/1 | COL | 8 | 0.062 | 32 | Synergy |
| | FOS | 4 | | 32 | |
| | COL | 8 | 0.093 | 32 | Synergy |
| | MIN | 2 | | 16 | |
| | COL | ND | ND | ND | ND |
| | TIG | ND | | | |
| | COL | 16 | 0.066 | 16 | Synergy |
| | PIP | 4 | | 256 | |
| 1NT6R(CCU) | COL | 32 | 0.375 | 8 | Synergy |
| | FOS | 256 | | 4 | |
| | COL | 8 | 0.062 | 32 | Synergy |
| | MIN | 4 | | 32 | |
| | COL | ND | ND | ND | ND |
| | TIG | ND | | | |
| | COL | 64 | 0.251 | 4 | Synergy |
| | PIP | 2 | | 512 | |

(*Continued*)

**Table 8.** (Continued)

| Isolates | Antibiotics | Combined MIC (µg/mL) | FICI | Fold reduction | Outcome |
|----------|-------------|----------------------|------|----------------|---------|
| 1TR5R | COL | 256 | 0.500 | 4 | Synergy |
| | FOS | 0.125 | | 4 | |
| | COL | ND | ND | ND | ND |
| | MIN | ND | | | |
| | COL | ND | ND | ND | ND |
| | TIG | ND | | | |
| | COL | 128 | 0.375 | 8 | Synergy |
| | PIP | 0.125 | | 4 | |

CCU, Cardiac Care Unit; MIC, Minimum inhibitory concentrations; COL, Colistin; FICI, Fractional inhibitory concentration index; FOS, Fosfomycin; MIN, Minocycline; PIP, Piperacillin; TIG, Tigecycline; ND, not determined.

colistin with levofloxacin was synergistic against 90.91% (10 of 11 isolates), while colistin and tigecycline combination were overall not synergistic. The results revealed that colistin in combination with fifteen other antibiotics could effectively inhibit colistin resistant isolates of *K. pneumoniae*. The results suggested that combination therapies could be further explore for the treatment of multidrug resistant pathogens.

## Conclusions

The present study combined colistin a last line drug for multidrug resistant Enterobacterales with 16 other antibiotics that are not effective against *K. pneumoniae*. The results revealed that colistin in combination with fifteen other antibiotics effectively inhibit colistin resistant isolates of *K. pneumoniae*. The results suggested that combination therapies could be further explore for the treatment of multidrug resistant pathogens.

## Supporting information

**S1 Table. Demographic data, clinical characteristics, and outcomes of the patients with colonization due to colistin-resistant and carbapenem-resistant *K. Pneumoniae*.**
(DOCX)

**S2 Table. Minimum Inhibitory Concentration (MIC) and Minimum Bactericidal Concentration (MBC) of colistin against *Klebsiella pneumoniae* clinical isolates.**
(DOCX)

**S3 Table. Synergistic effects of antibiotics and colistin combination against *Klebsiella pneumoniae* isolates from hospitalized patients.**
(DOCX)

## Author Contributions

**Conceptualization:** Julalak C. Ontong, Sarunyou Chusri.

**Data curation:** Julalak C. Ontong.

**Formal analysis:** Nwabor F. Ozioma.

**Funding acquisition:** Sarunyou Chusri.

**Investigation:** Julalak C. Ontong, Nwabor F. Ozioma.

**Methodology:** Julalak C. Ontong.

**Resources:** Supayang P. Voravuthikunchai.

**Supervision:** Supayang P. Voravuthikunchai, Sarunyou Chusri.

**Validation:** Supayang P. Voravuthikunchai, Sarunyou Chusri.

**Visualization:** Supayang P. Voravuthikunchai, Sarunyou Chusri.

**Writing – original draft:** Julalak C. Ontong, Nwabor F. Ozioma.

**Writing – review & editing:** Nwabor F. Ozioma, Sarunyou Chusri.

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
