## [Decision Letter · Decision Letter 0]

22 Oct 2020

PONE-D-20-27007

Synergistic antibacterial effects of colistin in combination with aminoglycoside, carbapenems, cephalosporins, fluoroquinolones, tetracyclines, fosfomycin, and piperacillin on multidrug resistant clinical isolates of Klebsiella pneumoniae

PLOS ONE

Dear Dr. chusri,

Thank you for submitting your manuscript to PLOS ONE. After careful consideration, we feel that it has merit but does not fully meet PLOS ONE’s publication criteria as it currently stands. Therefore, we invite you to submit a revised version of the manuscript that addresses the points raised during the review process.

A number of clarifications are needed in methodology and discussion.

We look forward to receiving your revised manuscript.

Kind regards,

Iddya Karunasagar

Academic Editor

PLOS ONE

Additional Editor Comments:

The reviewers have raised very useful points to improve the quality of manuscript. Clarifications are needed on CLSI guideline used and justifications for doing broth dilution for fosfomycin, MBC determination by checker board method rather than time kill method. Please respond to referee comments point by point.

Journal Requirements:

3. In the ethics statement in the manuscript and in the online submission form, please provide additional information about the patient records used in your retrospective study, including: a) the date range (month and year) during which patients' medical records were accessed; b) the date range (month and year) during which patients whose medical records were selected for this study sought treatment.

4. Thank you for stating the following in the Funding Section of your manuscript:

"This work is supported by Senior Research Scholar (Grant No. RTA6180006), the Thailand

Research Fund and Postdoctoral Fellowship, Prince of Songkla University."

"NO"

"NO"

7. Please upload a copy of Supporting Information Table S1 which you refer to in your text on page p8.

Reviewers' comments:

Reviewer's Responses to Questions

**Comments to the Author**

1. Is the manuscript technically sound, and do the data support the conclusions?

Reviewer #1: Yes

Reviewer #2: Yes

2. Has the statistical analysis been performed appropriately and rigorously? 

Reviewer #1: Yes

Reviewer #2: Yes

3. Have the authors made all data underlying the findings in their manuscript fully available?

Reviewer #1: Yes

Reviewer #2: Yes

4. Is the manuscript presented in an intelligible fashion and written in standard English?

Reviewer #1: Yes

Reviewer #2: Yes

5. Review Comments to the Author

Reviewer #1: The scientific work described in the manuscript is original, the research has been conducted using standard experimental techniques and references. The data has been analyzed and presented.The findings of the research can be applicable in the patient care.Some of the revisions suggested-

Introduction

Line 4 Infectious- infections

Line 6 Enterobacteriaceae- Enterobacteriaceae member

Line 10 bacteria isolates- bacterial isolates

Line 12- Gram-negative bacteria, colistin competitively displaces

Line 14- Colistin resistance in Enterobacteriaceae

What is the meaning of collection of antibiotics? clarify

Mention the limitation of this research work

Reviewer #2: Dear Authors

The paper describes the synergistic effect of colistin with antibiotics of different classes such as cephalosporins, carbapenems, fluoroquinalones, tetracyclins, fosfomycin, piperacillin tazobactam among others.

I request the authors to clarify on the following aspects

1. The breakpoints for colistin changed in 2020 both by CLSI and EUCAST to Intermediate and Resistant. Have the authors considering this change and has the data been interpreted in light of this change. If not, Why have the authors presented data with the old breakpoints

2. All the isolates studied from different centers seem to be colonizers like in the nasophyrynx, throat, rectal, ET tube and environmental cultures. Why were these organisms cultured and reported from these sites. What was the significance of these isolates clinically? Was an outbreak being studied? Or were these organisms in a special subset of patients. What is the clinical relevance of these isolates?

3. How was the MIC for fosfomycin done. The recommended method is agar dilution. Why was broth dilution used instead?

4. MBC results need to be explained better as Time Kill studies are the gold standard rather than the checkerboard method for synergy testing

5. Why were these FIC interpretations used. The reference to this is another article that finally leads to https://pubmed.ncbi.nlm.nih.gov/11168186/. The interpretation here seems to be different. So can the authors explain this?

6. What were the clinical outcomes in treating these patients? What were the antibiotics used?

7. The language needs to be looked into. Eg Ceftalozane tazobactam is wrongly spelled.

6. PLOS authors have the option to publish the peer review history of their article (what does this mean?). If published, this will include your full peer review and any attached files.

Reviewer #1: No

Reviewer #2: **Yes: **Dr. Anusha Rohit

---

## [Author Response · Author response to Decision Letter 0]

1 Nov 2020

Review Comments to the Author

Reviewer #1: 

The scientific work described in the manuscript is original, the research has been conducted using standard experimental techniques and references. The data has been analyzed and presented. The findings of the research can be applicable in the patient care. Some of the revisions suggested-

Introduction

Line 4 Infectious- infections

Corrected 

Line 6 Enterobacteriaceae- Enterobacteriaceae member

Corrected 

Line 10 bacteria isolates- bacterial isolates

Corrected 

Line 12- Gram-negative bacteria, colistin competitively displaces

Corrected 

Line 14- Colistin resistance in Enterobacteriaceae

Corrected 

What is the meaning of collection of antibiotics? Clarify

Clarification added. 

The resistant isolates were further tested against several groups of antibiotics, and the synergistic antibacterial effects of colistin combinations with other antibiotics were evaluated.

Mention the limitation of this research work

The study presents In vitro experimental antibacterial data from broth micro-dilution technique and the checkerboard assay, but did not monitor the time-kill.

Reviewer #2: 

Dear Authors

The paper describes the synergistic effect of colistin with antibiotics of different classes such as cephalosporins, carbapenems, fluoroquinalones, tetracyclins, fosfomycin, piperacillin tazobactam, among others.

I request the authors to clarify on the following aspects

1. The breakpoints for colistin changed in 2020 both by CLSI and EUCAST to Intermediate and Resistant. Have the authors considering this change and has the data been interpreted in light of this change. If not, Why have the authors presented data with the old breakpoints

YES, the new CLSI and EUCAST documents were considered in the interpretations.

The CLSI 2020 document presented a colistin break point of ≤ 2 as intermediate and ≥ 4 as resistant, whereas the EUCAST 2020 documents presents colistin breakpoints of ≤ 2 as susceptible and > 2 as resistant. Thus, we believe that colistin breakpoint as defined in the manuscript (≤ 2 as susceptible) was in line with the recommended interpretations.

2. All the isolates studied from different centers seem to be colonizers like in the nasophyrynx, throat, rectal, ET tube and environmental cultures. Why were these organisms cultured and reported from these sites. 

The study protocol included surveillance data of colonization of carbapenem-resistant Enterobacteriaceae in the patients who admitted in the intensive care units of each hospital. We obtained the specimens from the potential sites of colonization including throat, rectum, endotracheal tube (among patients with mechanical ventilator), gastric content (among the patients with oro/nasogastric tube) and hospital environment of the patients (bed sheet, pillowcase and bed rail).

What was the significance of these isolates clinically? Was an outbreak being studied? 

Or were these organisms in a special subset of patients. What is the clinical relevance of these isolates?

According to the rising prevalence of carbapenem-resistant Enterobacteriaceae in intensive care units (ICUs) of hospitals in Southern Thailand, a prospective epidemiological study was conducted to determine characteristics, risk factors and clinical outcomes of patients admitted in a network of hospitals including Songklanagarind Hospital, a university hospital with referral centres at 3 tertiary care hospital (Yala, Trang, and Hat Yai Hospital) and 5 provincial hospitals (Pattani, Narathiwat, Phatthalung, Satun, and Songkhla Hospital). We randomly selected only one isolates of K. pneumoniae from patients with multiple sites of colonization and only one isolate for the patients with multiple sites of environmental contamination.

3. How was the MIC for fosfomycin done. The recommended method is agar dilution. Why was broth dilution used instead?

Although the agar dilution method is recommended as reference method for the determination of Fosfomycin MICs, using the reference agar dilution method in a checkerboard analysis is practically difficult. Thus, the broth microdilution with glucose-6-phosphate (G-6-P) was used [28].

28. Flamm RK, Rhomberg PR, Lindley JM, Sweeney K, Ellis-Grosse E, Shortridge D. Evaluation of the bactericidal activity of fosfomycin in combination with selected antimicrobial comparison agents tested against gram-negative bacterial strains by using time-kill curves. Antimicrobial Agents and Chemotherapy. 2019;63(5).

4. MBC results need to be explained better as Time Kill studies are the gold standard rather than the checkerboard method for synergy testing

MBC methodology added.

To determine the MBC, MIC and supra-MIC dilutions were spotted on an agar plate and incubated overnight at 37 °C. Bacterial growth was observed, and MBC was defined as the lowest concentration that showed no visible bacterial regrowth. 

The Time-kill kinetics were not monitored in the study. The checkerboard assay was not used for MBC determination, but for determination of drug synergy.

5. Why were these FIC interpretations used. The reference to this is another article that finally leads to https://pubmed.ncbi.nlm.nih.gov/11168186/. The interpretation here seems to be different. So can the authors explain this?

The FICs were interpretated as.

FICI ≤ 0.5 – synergism; 0.5 < FICI < 1– additive; 1 ≤ FICI < 2 – indifference; and FICI ≥ 2 – antagonism, which are in agreement with the cited article and your reference article

https://pubmed.ncbi.nlm.nih.gov/11168186/.

6. What were the clinical outcomes in treating these patients? What were the antibiotics used?

The requested information has been provided in Table S1

7. The language needs to be looked into. Eg Ceftalozane tazobactam is wrongly spelled.

The entire manuscript has been re-read and revised.

---

## [Decision Letter · Decision Letter 1]

7 Dec 2020

PONE-D-20-27007R1

Synergistic antibacterial effects of colistin in combination with aminoglycoside, carbapenems, cephalosporins, fluoroquinolones, tetracyclines, fosfomycin, and piperacillin on multidrug resistant clinical isolates of Klebsiella pneumoniae

PLOS ONE

Dear Dr. chusri,

Thank you for submitting your manuscript to PLOS ONE. After careful consideration, we feel that it has merit but does not fully meet PLOS ONE’s publication criteria as it currently stands. Therefore, we invite you to submit a revised version of the manuscript that addresses the points raised during the review process.

We look forward to receiving your revised manuscript.

Kind regards,

Iddya Karunasagar

Academic Editor

PLOS ONE

Additional Editor Comments (if provided):

The authors need to clarify regarding the isolates and provide justification to call them "clinical isolates". Also treatment with antibiotics for colonisers should be justified.

Reviewers' comments:

Reviewer's Responses to Questions

**Comments to the Author**

1. If the authors have adequately addressed your comments raised in a previous round of review and you feel that this manuscript is now acceptable for publication, you may indicate that here to bypass the “Comments to the Author” section, enter your conflict of interest statement in the “Confidential to Editor” section, and submit your "Accept" recommendation.

Reviewer #1: All comments have been addressed

Reviewer #2: (No Response)

2. Is the manuscript technically sound, and do the data support the conclusions?

Reviewer #1: Yes

Reviewer #2: Yes

3. Has the statistical analysis been performed appropriately and rigorously? 

Reviewer #1: Yes

Reviewer #2: Yes

4. Have the authors made all data underlying the findings in their manuscript fully available?

Reviewer #1: Yes

Reviewer #2: Yes

5. Is the manuscript presented in an intelligible fashion and written in standard English?

Reviewer #1: Yes

Reviewer #2: Yes

6. Review Comments to the Author

Reviewer #1: All the comments have been addressed and the whole manuscript is revised and rewritten. This work will be an addition to the medical literature.

Reviewer #2: Dear Authors

All comments have been addressed but the following still needs to be addressed.

The answer to the reviewer explains that the isolates was for surveillance of the throat, rectum etc. But the manuscript still calls them clinical isolates. The clarity of the two is important in the fact that colonizers are not treated with antibiotics where as pathogens are. Hence this distinction has to be made more distinctly.

It is also important to mention Enterobacterales as the order rather than Enterobacteriaceae as a family is accepted nomenclature in the present times.

The other queries have been answered adequately.

7. PLOS authors have the option to publish the peer review history of their article (what does this mean?). If published, this will include your full peer review and any attached files.

Reviewer #1: No

Reviewer #2: No

---

## [Author Response · Author response to Decision Letter 1]

11 Dec 2020

Additional Editor Comments

The authors need to clarify regarding the isolates and provide justification to call them "clinical isolates". Also treatment with antibiotics for colonisers should be justified.

Reviewer #2: Dear Authors

The answer to the reviewer explains that the isolates was for surveillance of the throat, rectum etc. But the manuscript still calls them clinical isolates. 

Response: The isolates used for this study were obtained from patients receiving treatment in tertiary hospitals. 

Furthermore, it is important to mention that colonizers can become opportunistic pathogens especially in immunocompromised individuals. 

The clarity of the two is important in the fact that colonizers are not treated with antibiotics whereas pathogens are. Hence this distinction has to be made more distinctly.

Response: In response to your queries, we have replaced the term clinical isolates with Klebsiella pneumonia isolates. We also want to note that the antibiotics treatments were administered prior to inclusion into the surveillance study.

It is also important to mention Enterobacterales as the order rather than Enterobacteriaceae as a family is accepted nomenclature in the present times.

Corrected throughout the manuscript.

---

## [Editor Report · Decision Letter 2]

15 Dec 2020

Synergistic antibacterial effects of colistin in combination with aminoglycoside, carbapenems, cephalosporins, fluoroquinolones, tetracyclines, fosfomycin, and piperacillin on multidrug resistant Klebsiella pneumoniae isolates

PONE-D-20-27007R2

Dear Dr. chusri,

We’re pleased to inform you that your manuscript has been judged scientifically suitable for publication and will be formally accepted for publication once it meets all outstanding technical requirements.

Kind regards,

Iddya Karunasagar

Academic Editor

PLOS ONE

Additional Editor Comments (optional):

All reviewer comments addressed satisfactorily
---

## [Editor Report · Acceptance letter]

18 Dec 2020

PONE-D-20-27007R2 

Synergistic antibacterial effects of colistin in combination with aminoglycoside, carbapenems, cephalosporins, fluoroquinolones, tetracyclines, fosfomycin, and piperacillin on multidrug resistant *Klebsiella pneumoniae* isolates 

Dear Dr. chusri:

I'm pleased to inform you that your manuscript has been deemed suitable for publication in PLOS ONE. Congratulations! Your manuscript is now with our production department. 

Kind regards, 

on behalf of

Dr. Iddya Karunasagar 

Academic Editor

PLOS ONE